

# Clustering U-14 Portuguese regional team football players by lower limb strength, power, dynamic balance, speed and change of direction: understanding the field position factor

Tatiana Sampaio[1,2], Daniel Marinho[2,3], José Eduardo Teixeira[1,2,4], João Oliveira[1,2] and Jorge Morais[1,2]

[1] Instituto Politécnico de Bragança, Bragança, Portugal
[2] Research Center in Sports Sciences, Health Sciences and Human Development (CIDESD), Covilhã, Portugal
[3] University of Beira Interior, Covilhã, Portugal
[4] Instituto Politécnico da Guarda, Guarda, Portugal

Corresponding author
Jorge Morais,
morais.jorgestrela@gmail.com

## ABSTRACT

**Objectives:** The aim of this study was to cluster U-14 Portuguese regional team football players based on variables related to lower limb strength and power, dynamic balance, linear sprint, and change of direction.

**Method:** The sample consisted of 22 young male soccer players (13.83 ± 0.44 years). A set of variables related to lower limb strength and power, dynamic balance, linear sprint, and change of direction was measured.

**Results:** Overall, a non-significant field position was observed. The countermovement jump ($p < 0.001$, $\eta^2 = 0.73$), squat jump ($p < 0.001$), and 30 m linear sprint ($p = 0.001$) were the main variables responsible for establishing the clusters. Cluster 1 was characterized by a high slalom, *i.e.*, it took longer to complete the test (speed and change of direction) and a low composite score in the dynamic balance. Cluster 2 was characterized by high squat jump, countermovement jump, and reactive strength ratio (lower limb strength and power). Cluster 3 was characterized by low squat jump and countermovement jump (lower limb strength and power), and a high 30 m sprint, *i.e.*, it took longer to complete the test (speed and change of direction). Based on the results of the territorial map, the clusters consisted of: (i) cluster 1: two forwards, one midfielder, and five defenders; (ii) cluster 2: three forwards, two midfielders, and two defenders, and; (iii) cluster 3: four midfielders and three defenders.

**Conclusions:** Players from different field positions composed the three clusters. There was no cluster composed exclusively of players of the same field position. The development of individualized and specific enrichment training programs should consider a cluster analysis, as the positional effect can be overlooked.

## INTRODUCTION

Football is a team sport where the performance depends on several domains, such as tactical (*Memmert, Lemmink & Sampaio, 2017*), technical (*Yi et al., 2019*), physical (*Morgans et al., 2022*), nutritional (*Carter et al., 2022*), and psychological (*Coutinho et al., 2022*). Notwithstanding, the assessment of physical traits in football settings has been extensively studied (*Emmonds et al., 2019*; *Morris et al., 2018*). Physical field-based measures make it possible to profile the players in order to better understand their physical condition as well as their strengths and handicaps (*Morris et al., 2018*).

A football game is typically characterized by intermittent efforts with periods of high-intensity, such as sprints, accelerations, decelerations, jumps, and impacts driven by the lower limbs (*Teixeira et al., 2021a*). Therefore, football players must have high levels of maximum strength, functional capacity, speed, and explosive power in their lower limbs to improve their physical performance (*Bennett et al., 2019*; *Sanchez-Sanchez et al., 2021*). Linear sprint and change of direction tests have been widely applied to measure physical performance (*Köklü et al., 2015*; *Trecroci et al., 2020*). Moreover, tests that measure the players' lower limb strength and power (squat jump—SJ; countermovement jump—CMJ; reactive strength ratio—RSR) allow the measurement of lower limb strength and power (*Markovic et al., 2004*). Concerning the dynamic balance of the lower trunk, high reliability values have been reported for the Y-balance test for youth football players (*Plisky et al., 2009*). Indeed, the most recent evidence on this topic indicates that linear sprint, jump performance, and dynamic balance are determinants of change of direction tests (*Falces-Prieto et al., 2022*). Therefore, it can be stated that these parameters are interconnected and represent the performance of a football player. Additionally, the practical applications of field-based testing can be extended to dose-response management (*Ellis et al., 2021*), injury prevention (*Gabbett, 2016*), and training planning (*Clemente et al., 2022*), in addition to fitness assessment.

Overall, there are at least three main field positions (besides the goalkeeper), *i.e.*, defenders, midfielders, and forwards. The literature on football players also provides information on the players' physical traits by field position (*Morgans et al., 2022*). Indeed, the positional role can be highly dependent on physical qualities or motor skills, therefore defining the physiological and biomechanical profiles is a critical issue in football (*Teixeira et al., 2021b*; *Tomáš et al., 2014*). If, on the one hand, there is significant information on elite adults or young players (*Abade et al., 2014*), there is scarce information regarding young players at regional level who can participate in competitions at national level. At least in elite youth football, discriminatory power has been reported for lower limb strength, power, dynamic balance, speed, and performance on change of direction tests to group players into playing positions (*Emmonds et al., 2019*; *Morris et al., 2018*; *Tomáš et al., 2014*). In particular, it was reported that the percentage of U-14 players who were correctly classified into positional groups was slightly lower when the analysis was performed on the overall team (*Silva et al., 2010*). Although differences in physical performance between sub-elite and elite football players have been reported, little is known

about lower limb strength, power, dynamic balance, speed, and change of direction tests in sub-elite football training (*Trecroci et al., 2018*, *2019*).

Few studies have analyzed physical performance and training load monitoring in sub-elite youth football (*Teixeira et al., 2021b*, *2022*; *Trecroci et al., 2019*). These reported a small positional role effect as opposed to elite football contexts. Moreover, as far as is known, there is no information about how these variables can determine the position of each young player in the field. Cluster analysis is a viable procedure for assessing individual trends within an overall sample (*Rein et al., 2010*). Such multivariate data analysis can be employed to detect patterns within data sets by grouping together subjects who share several common characteristics but are very different from one another (*Rein et al., 2010*). Therefore, the aim of this study was to cluster U-14 Portuguese regional team football players based on variables related to lower limb strength and power, dynamic balance, linear sprint, and change of direction variables. It was hypothesized that those with the best scores on the sprint and change of direction variables would be those with the highest levels of strength and power, and players would also be clustered primarily by their field position.

## MATERIALS AND METHODS

### Participants

The sample consisted of 22 young male soccer players ($13.83 \pm 0.44$ years; $55.57 \pm 7.22$ kg of body mass; $168.68 \pm 5.17$ cm of height; $-0.98 \pm 0.82$ years of maturity offset) including 10 defenders, seven midfielders, and five forwards. The players regularly participated in the regional championship. For this study, they were recruited from a regional squad that selected the best players from a regional championship to participate in a national competition that brought together a set of regional teams. They were considered Tier 2 athletes (*McKay et al., 2021*). Data collection was performed 1 week after the last regional championship match. From the selection phase until the beginning of the national championship, they had three training sessions per week for 2 weeks. In order to be included in the assessments, players had to be completely free of pain at the time of the study. They would be excluded if they were receiving medical attention at the time or indicated any pain during the Y-balance test (please report to methods section). Parents or guardians and players signed an informed consent form. All procedures were in accordance with the Declaration of Helsinki regarding human research, and the Polytechnic Ethics Board approved the research (No. 127/2023).

### Anthropometrics and maturity offset

Body mass (in kg) was measured on an electronic scale (MC 780-P; Tanita, Tokyo, Japan) with minimal clothing. Height (in cm) was measured using an electronic stadiometer (Seca 242; Seca, Hamburg, Germany). The maturity offset was calculated as suggested elsewhere (*Mirwald et al., 2002*). This represents the years an athlete is away from peak height velocity. If the offset is negative, it means that the athlete has not yet reached the peak height velocity. A positive offset indicates that the peak height velocity has already occurred.

## Linear sprint and change of direction tests

Before data collection, players performed a standardized warm-up based on muscle activation monitored by their coach. The 30 m linear sprint, slalom, and forward-back-forward tests were chosen as performance variables and collected on a synthetic grass pitch (*Sporis et al., 2010*). The slalom and forward-back-forward tests are considered change of direction tests based on the agility needed to perform them with great scores (*Sporis et al., 2010*). Each player performed each test three times at maximum speed with a 10-min recovery time in between (*Köklü et al., 2015*). Subsequently, the fastest time was used for further analysis.

The 30 m linear sprint test consisted of running this distance in a straight line in the shortest time. For the slalom test, the players started the protocol behind the starting point. The first cone was placed 1 m apart from the starting line and the other five cones were placed 2 m apart, with the last one being placed at 11 m from the first. The test consisted of running from the starting line, changing the direction from right to left (slalom) until the last cone and coming back (also performing changes in direction) until reaching the starting line (*Sporis et al., 2010*). For the forward-back-forward test, the players were instructed to run 9 m in a straight line. When they reached that mark, they had to run backward to the 6 m mark, followed by a forward run to the 12 m mark. When reaching this mark, they had to run backward to the 9 m mark, followed by a straight line run to the 21 m mark (*Sporis et al., 2010*). Figure 1 shows the schematics of the three tests.

All tests were timed with Microgate Witty photocells (Microgate, Bolzano, Italy). The photocells were placed 0.4 m above the ground to minimize the effect of the hand swing when passing through the gate. They were activated when crossed. The players were instructed to start their attempts just before the first photocell (0.1 m), and the timer started at their very first movement after crossing the photocells. Participants were instructed to start whenever they wanted in order to have a faster and more reliable start (*Nuell et al., 2020*). For the 30 m sprint and forward-back-forward tests, two set of gates were used (one for the start and another for the finish). For the slalom test, one set of gates were used (the same set worked as start and finish).

## Dynamic balance of the lower trunk

The Y-balance test kit was used to measure the players' dynamic balance of the lower trunk (*Plisky et al., 2009*). Before the measurements, the players familiarized with the protocol and tested it. The protocol consists of the participant reaching with one foot in the anterior, posteromedial, and posterolateral directions while supporting the other foot on a centralized stance platform. The test is performed barefoot with both lower limbs and repeated three times with each lower limb (*Plisky et al., 2009*). The players placed their stance-foot toes immediately behind the start line. Afterward, they were instructed to reach as far as they could while maintaining their balance. The anterior, posteromedial, and posterolateral directions must be performed consecutively with a loss of balance that promotes the players touching the ground. If the players failed any attempt (anterior, posteromedial, and posterolateral), they were asked to perform the trial again from the beginning. The Y-balance test was supervised by an expert evaluator. The sum of the three

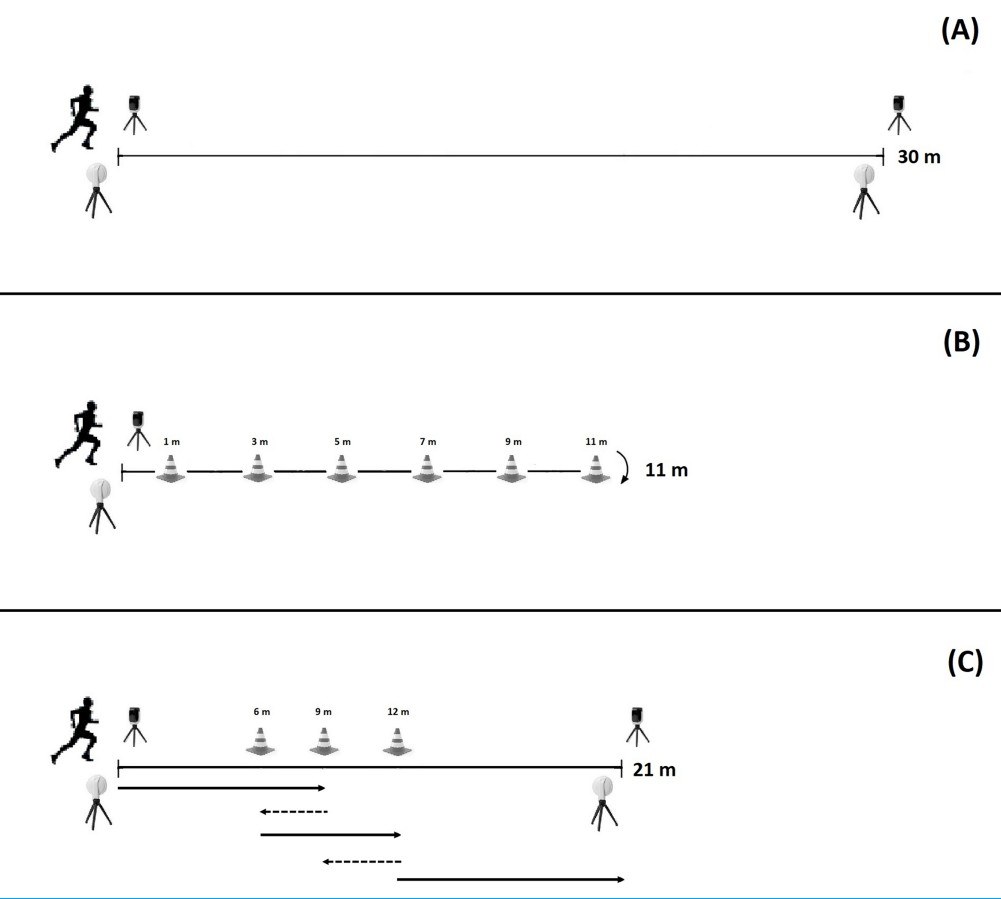

**Figure 1 Tests schematics.** (A) The 30 m sprint test; straight line sprinting over 30 m. (B) Slalom test; changing of direction test by performing a slalom around cones and coming back also performing the slalom. (C) Forward-back-forward test; changing of direction test by sprinting forward and backward in specific marks. Solid lines indicate forward sprinting and dash lines backward sprinting.

normalized reach distances was then averaged and multiplied by 100 to generate a composite score (CS). The absolute reach difference between lower limbs was calculated to assess reach symmetry (*Plisky et al., 2009*).

## Lower limbs' strength and power

The squat jump (SJ), countermovement jump (CMJ), and reactive strength ratio (RSR) were used as indicators of lower limb strength and power. Each player executed each test three times and the recovery time between trials was 10 min. The best trial was used for further analysis. For the SJ, players were instructed to jump from a semi-squat position with no countermovement. Specifically, they started from a standing position with their hands on their hips and were instructed to bend their knees at an angle of 90° approximately. After holding that position for 3 s, they were instructed to jump as high as possible without performing any countermovement before the execution of the jump (*Glatthorn et al., 2011*). For the CMJ, the players started from the same upright standing position with their hands on their hips. Afterward, they were instructed to perform a

countermovement with the lower limbs (*i.e.*, knee flexion to about 90°) before performing a vertical jump (*Markovic et al., 2004*). They were instructed to land on the same starting point and with the lower limbs straight to avoid knee flexion, which could lead to erroneous results. For the RSR, the players performed a drop jump from a stable platform of 45 cm of height (*Werstein & Lund, 2012*). The players were instructed to let themselves fall with their lower limbs in an erect position, and immediately upon contact with the ground, jump vertically as high as they could. The RSR was calculated as the ratio between flight and contact time (*Healy, Kenny & Harrison, 2016*). All tests were measured with an Optojump system (Microgate, Bolzano, Italy) with the bars separated by 1 m (*Glatthorn et al., 2011*). The validity and reliability of this equipment have already been confirmed (*Glatthorn et al., 2011*).

**Statistical analysis**

The normality assumption was analyzed with the Shapiro–Wilk test which revealed a normal distribution. The mean plus one standard deviation was calculated as descriptive statistics. The one-way ANOVA was used to analyze the players' variance according to their position in the field (*i.e.*, defenders, midfielders, and forwards). The total eta square ($\eta^2$) was selected as the effect size index, and deemed as: (i) without effect if $0 < \eta^2 < 0.04$; (ii) minimum if $0.04 < \eta^2 < 0.25$; (iii) moderate if $0.25 < \eta^2 < 0.64$ and; (iv) strong if $\eta^2 > 0.64$ (*Ferguson, 2009*). The level of significance was set at $\alpha = 0.05$. If a significant variance was verified, the Bonferroni correction was used to verify the differences between pairwise ($p < 0.017$). Cohen's d was used to estimate the standardized effect sizes, and was deemed as: (i) trivial if $0 \leq d < 0.20$; (ii) small if $0.20 \leq d < 0.60$; (iii) moderate if $0.60 \leq d < 1.20$; (iv) large if $1.20 \leq d < 2.00$; (v) very large if $2.00 \leq d < 4.00$; (vi) nearly distinct if $d \geq 4.00$ (*Hopkins, 2019*).

Cluster modeling was performed based on the k-means approach (non-hierarchical). This allows the definition of a number of clusters to be used in advance. The k-means defines a centroid (*i.e.*, the mean of a group of points/subjects) based on their similarities (*Rein et al., 2010*). Standardized z-scores were used to ensure a coherent comparison of data sets with different magnitudes and/or units. The elbow method was used to understand the number of clusters to be retained for analysis. For this, only the variables related to lower limb strength and power (SJ, CMJ, and RSR), dynamic balance (CS), and speed and change of direction tests (30 m sprint, slalom, and forward-back-forward) were included. The one-way ANOVA was used to identify the main determinants responsible for establishing the clusters ($p < 0.05$). The total eta square ($\eta^2$) was selected as effect size index and deemed as mentioned before. Afterward, discriminant analysis (stepwise method) was performed to validate the clusters. All statistical analysis were performed with the IBM SPSS statistics program (version 26.0; IBM Inc., Chicago, IL, USA).

## RESULTS

Table 1 presents the players' demographics, variables measured by field position, and all players plotted together. Overall, for all sets of variables (demographics, lower limb strength and power, dynamic balance, linear sprint and change of direction tests), a

**Table 1 Descriptive statistics.**

| | | Mean ± SD | | | | |
|---|---|---|---|---|---|---|
| | All (N = 22) | Defenders (N = 10) | Midfielders (N = 7) | Forwards (N = 5) | F-ratio (p) | $\eta^2$ |
| **Demographics** | | | | | | |
| Age (years) | 13.83 ± 0.44 | 13.83 ± 0.55 | 13.86 ± 0.37 | 13.79 ± 0.37 | 0.04 (0.996) | 0.00 |
| Body mass (kg) | 55.57 ± 7.22 | 57.95 ± 5.55 | 56.50 ± 7.59 | 49.52 ± 7.52 | 2.75 (0.089) | 0.23 |
| Height (cm) | 168.68 ± 5.17 | 170.60 ± 4.95 | 168.29 ±3.40 | 165.40 ± 6.69 | 1.86 (0.183) | 0.16 |
| Maturity offset (years) | −0.98 ± 0.82 | −0.86 ± 1.01 | −0.86 ± 0.55 | −1.38 ± 0.70 | 0.77 (0.478) | 0.08 |
| **Lower limbs strength and power** | | | | | | |
| SJ (cm) | 28.72 ± 3.91 | 28.29 ± 4.69 | 28.11 ± 3.74 | 30.44 ± 2.27 | 0.60 (0.557) | 0.06 |
| CMJ (cm) | 29.64 ± 4.06 | 28.48 ± 4.07 | 29.40 ± 4.44 | 32.28 ± 2.72 | 1.56 (0.237) | 0.14 |
| RSR (m/s) | 0.81 ± 0.28 | 0.77 ± 0.27 | 0.72 ± 0.31 | 1.06 ± 0.22 | 2.03 (0.158) | 0.18 |
| **Dynamic balance** | | | | | | |
| CS (%) | 94.44 ± 11.98 | 93.72 ± 8.68 | 97.770 ± 10.61 | 91.31 ± 19.57 | 0.42 (0.661) | 0.04 |
| Anterior absolute difference (cm) | 3.12 ± 1.99 | 3.73 ± 2.04 | 3.00 ± 2.26 | 2.07 ± 1.14 | 1.22 (0.317) | 0.11 |
| Anterior relative difference (%) | 3.61 ± 2.46 | 4.16 ± 2.49 | 3.66 ± 2.78 | 2.42 ± 1.93 | 0.82 (0.455) | 0.08 |
| Postero-lateral absolute difference (cm) | 4.03 ± 3.42 | 3.73 ± 3.08 | 4.05 ± 4.66 | 4.60 ± 2.62 | 0.10 (0.907) | 0.01 |
| Postero-lateral relative difference (%) | 4.26 ± 3.88 | 4.30 ± 3.94 | 3.95 ± 4.66 | 4.62 ± 3.33 | 0.04 (0.960) | 0.00 |
| Postero-medial absolute difference (cm) | 5.10 ± 3.80 | 7.46 ± 4.23 | 2.38 ± 1.42 | 4.20 ± 2.15 | 5.53 (0.013) | 0.37 |
| Postero-medial relative difference (%) | 5.19 ± 4.36 | 8.13 ± 4.81 | 2.10 ± 1.52 | 3.61 ± 1.67 | 6.77 (0.006) | 0.42 |
| **Speed and change of direction** | | | | | | |
| 30 m Sprint (s) | 4.96 ± 0.24 | 4.91 ± 0.17 | 5.09 ± 0.30 | 4.87 ± 0.23 | 1.77 (0.197) | 0.16 |
| Slalom (s) | 6.70 ± 0.57 | 6.85 ± 0.72 | 6.75 ± 0.27 | 6.32 ± 0.42 | 1.58 (0.232) | 0.14 |
| Forward-back-forward (s) | 10.65 ± 1.61 | 10.94 ± 1.86 | 10.41 ± 1.36 | 10.39 ± 1.65 | 0.28 (0.758) | 0.03 |

Note:
SJ, squat jump; CMJ, countermovement jump; CS, composite score.

non-significant field position was observed. A significant variance with a moderate effect size was only observed in dynamic balance for the postero-medial difference variable (absolute: F = 5.53, p = 0.013, $\eta^2$ = 0.37; relative: F = 6.77, p = 0.006, $\eta^2$ = 0.42). The pairwise comparison revealed significant differences with large effect sizes between the defenders and midfielders for the postero-medial difference variable (absolute: mean difference = 5.08, p = 0.013, d = 1.61; relative: mean difference = 6.04, p = 0.007, d = 1.69).

The elbow method was used to test several cluster solutions (from 2 to 9; *i.e.*, 2 < k < 9). The three-cluster solution (k = 3) presented the highest power with smaller gains after the fourth cluster. Table 2 presents the descriptive and inferential data of variables related to the lower limb strength and power, dynamic balance, and linear sprint, and change of direction tests after the cluster modeling. The CMJ (F = 25.18, p < 0.001, $\eta^2$ = 0.73), SJ (F = 14.40, p < 0.001, $\eta^2$ = 0.60), and 30 m linear sprint (F = 10.12, p = 0.001, $\eta^2$ = 0.52) were the main variables responsible for establishing the clusters. Cluster 1 was characterized by a high slalom, *i.e.*, it took longer to complete the test, and a low CS (dynamic balance). Cluster 2 was characterized by high SJ, CMJ, and RSR (lower limb strength and power). Cluster 3 was characterized by low SJ and CMJ (lower limb strength
**Table 2 Clustering statistics.**

| | Cluster 1 (N = 8) | | Cluster 2 (N = 7) | | Cluster 3 (N = 7) | | F-ratio (p) | η² |
|---|---|---|---|---|---|---|---|---|
| | Mean ± SD | z-score | Mean ± SD | z-score | Mean ± SD | z-score | | |
| SJ (cm) | 28.31 ± 2.48 | −0.1002 | 32.66 ± 2.33 | 1.0605 | 25.26 ± 2.95 | −0.9164 | 14.40 (<0.001) | 0.60 |
| CMJ (cm) | 29.69 ± 2.44 | −0.0053 | 33.84 ± 2.07 | 1.0589 | 25.37 ± 2.14 | −1.1097 | 25.18 (<0.001) | 0.73 |
| RSR (m/s) | 0.79 ± 0.20 | −0.1503 | 1.08 ± 0.19 | 0.8993 | 0.58 ± 0.24 | −0.8894 | 9.90 (0.001) | 0.51 |
| CS (%) | 87.80 ± 11.89 | −0.6376 | 102.40 ± 8.96 | 0.4744 | 94.07 ± 11.15 | −0.1599 | 3.41 (0.054) | 0.26 |
| 30 m Sprint (s) | 4.80 ± 0.15 | −0.7526 | 4.90 ± 0.12 | −0.3096 | 5.20 ± 0.24 | 0.9057 | 10.12 (0.001) | 0.52 |
| Slalom (s) | 6.92 ± 0.59 | 0.3690 | 6.64 ± 0.51 | −0.1382 | 6.51 ± 0.60 | −0.3720 | 1.02 (0.381) | 0.10 |
| Forward-back-forward (s) | 10.98 ± 1.56 | 0.0833 | 10.80 ± 1.78 | −0.0227 | 10.12 ± 1.62 | −0.4274 | 0.59 (0.588) | 0.05 |

Note:
SJ, squat jump; CMJ, countermovement jump; CS, composite score; $p$, significance value; $\eta^2$, eta square (effect size index).

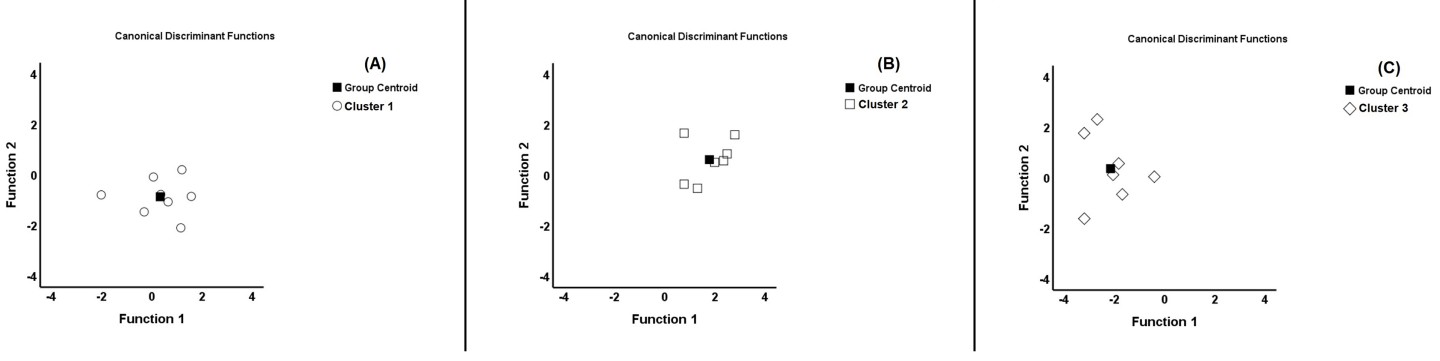

**Figure 2 Territorial map.** Territorial map for each cluster. (A) Cluster 1; (B) cluster 2; (C) cluster 3. ■, Centroid: ○, cluster 1 membership; □, cluster 2 membership; ◊, cluster 3 membership.

and power), and a high 30 m linear sprint, *i.e.*, it took longer to complete the test. The clusters consisted of: (i) cluster 1: two forwards, one midfielder, and five defenders; (ii) cluster 2: three forwards, two midfielders, and two defenders, and; (iii) cluster 3: four midfielders and three defenders. Thus, players from different field positions were part of the three clusters. There was no cluster that exclusively consisted of players of the same field position in sub-elite football players.

Figure 2 shows the stepwise discriminant analysis as qualitative assessment. This extracted two functions including the CMJ and the 30 m linear sprint. Function 1 was mainly defined by the CMJ, explaining 85.2% of the variance ($\Lambda = 0.169$, $X^2 = 32.840$, $p < 0.001$). Function 2 was mainly defined by the 30 m linear sprint, explaining 14.8% of the variance ($\Lambda = 0.664$, $X^2 = 7.583$, $p = 0.006$). Based on the results of the territorial map, the discriminant analysis showed good compactness/separation with a correct classification of the original groups (77.3%) and a correct classification of cross-validated groups (77.3%) (Fig. 2). The classification functions are as follows:

$$Cluster\ 1 = 8.167 \cdot CMJ + 173.518 \cdot 30\ m\ linear\ sprint - 538.599 \tag{1}$$

$$Cluster\ 2 = 9.068 \cdot CMJ + 178.857 \cdot 30\ m\ linear\ sprint - 593.124 \qquad (2)$$

$$Cluster\ 3 = 7.448 \cdot CMJ + 184.999 \cdot 30\ m\ linear\ sprint - 576.186 \qquad (3)$$

These functions make it possible to understand in which cluster (based on the CMJ and 30 m linear sprint tests) other players would be included.

## DISCUSSION

The aim of this study was to cluster U-14 Portuguese regional team football players based on variables related to lower limb strength and power, dynamic balance, linear sprint and change of direction variables. The main findings indicate that, from all the measured variables, only the postero-medial difference (dynamic balance) revealed a significant field position effect. Moreover, cluster analysis revealed that no cluster consisted exclusively of players of the same field position. Thus, it seems that lower limb strength and power, linear sprint and change of direction tests, as well as dynamic balance did not discriminate the field position in young regional level players.

In football research, studies aim to provide detailed information about the players' characteristics based on their field position (*Teixeira et al., 2021b*, *2022*; *Trecroci et al., 2019*). It is well-known that, depending on their position, players (*i.e.*, defenders, midfielders, or forwards) have different indexes of lower limb strength and power (*Gissis et al., 2006*; *Portes et al., 2015*), linear sprint and change of direction tests (*Portes et al., 2015*; *Rebelo et al., 2012*), and dynamic balance (*González-Fernández et al., 2022*). Usually, defenders have a higher flexibility and a lower limb strength and muscle power than midfielders and forwards (*Gissis et al., 2006*; *Portes et al., 2015*). Also, elite football players presented a better agility performance than non-elite players in all positional roles (*Rebelo et al., 2012*). Previous comparisons between field positions revealed that dynamic balance was lower in center-backs than wingers and forwards (*González-Fernández et al., 2022*). Additionally, football players with the highest speed and change of direction tests values also have equally better lower limb strength and power (*Portes et al., 2015*; *Rebelo et al., 2012*; *Trecroci et al., 2016*).

On the other hand, the present findings indicated a non-significant field position in most of the measured variables. Only a significant variance with a moderate effect size was found in dynamic balance for the poster-medial difference. Previous research has reported an isometric hamstring to quadriceps ratio differences between age groups (*Peek et al., 2018*). This positional difference was already reported in other team sports in the lower quarter Y-balance test (*Krombholz et al., 2023*). However, the current data indicate only a significant variance with a moderate effect size in dynamic balance for postero-medial difference. Consequently, the players evaluated in the present study do not seem to demonstrate the patterns seen in other young players (but of greater expertise).

The literature shows that pubertal development is strongly related to performance in sprint ability (*Perroni et al., 2018*). No significant effects of maturity offset were found between players of different field position. Nonetheless, the forwards in the present study were the ones with the lowest maturity offset scores. Controversially, they were the ones who

presented the best scores in the lower limb strength and power, linear sprint, and change of direction tests. That is, they were farther away from reaching the peak height velocity compared to defenders and midfielders, but they were more powerful and faster than their counterparts.

Cluster modeling was used to understand how these players could be gathered in groups with similar characteristics. The CMJ, SJ, and 30 m linear sprint were the main variables responsible for establishing the clusters. This indicates that the lower limb strength and power and the speed achieved in a straight line were the main variables discriminating the three clusters. This highlights the importance of developing young players' strength and power to cope with the effort and task demands of football training and matches (*Clemente et al., 2022*; *Teixeira et al., 2021a*). When differentiating the clusters assembled in this study, players in cluster 1 were characterized by a poor performance in the slalom test and a low CS (dynamic balance). Players in cluster 2 were characterized by high SJ, CMJ, and RSR (*i.e.*, large indexes of lower limb strength and power). Players in cluster 3 were characterized by low SJ and CMJ (small indexes of lower limb strength and power), and a poor performance in the 30 m linear sprint. Although players grouped in cluster 2 had the highest values of strength and power, they only performed better in the linear sprint test. Therefore, it appears that the strength and power provided may not have a direct influence on the change of direction test tasks, at least in these players. Indeed, the literature reported that agility and change of direction tests are autonomous skills that depend on other factors, such as reaction time, decision-making and coordination (*Young, Rayner & Talpey, 2021*). Also, linear and curvilinear movements have different profiles of physical qualities contributions (*Portes et al., 2015*; *Rebelo et al., 2012*).

Hence, the cluster analysis seems to have confirmed the non-significant effect of field position, as there was no cluster that was exclusively composed of players of the same field position. All clusters have several players of different positions, which means that the training design of these players should be planned with similar characteristics rather than by their field position. This means that, more than differentiating by field position, it is important to define the biomechanical profiles that will later allow the development of specific training programs for individualized enrichment (*Coutinho et al., 2018*; *Rudd et al., 2020*). Typically, jumping exercises should be introduced into the training microcycle with an initial strength training exercise. Furthermore, if one considers the results of the territorial map, players from different field positions were clustered into different groups. The current findings contradict previous research that reported positional differences between playing positions (*Emmonds et al., 2019*; *Morris et al., 2018*; *Tomáš et al., 2014*). The average values of the aforementioned variables are also lower compared to other studies (*Emmonds et al., 2019*; *Morris et al., 2018*; *Tomáš et al., 2014*), thus the level expertise seems to have a high influence on the variables assessed in this study (regional level players). Moreover, a recent study by *Perroni et al. (2023)* used factorial analysis to understand the physical performance in young football players according to their training status. The battery tests included endurance tests and repeated sprint ability, and also linear sprints and lower limb strength and power tests. The training status was indicated based on testosterone and cortisol saliva concentrations. Biological

maturation was also measured. The authors observed that a three-component model was the one that best explained (57% of the total variance) the young players' profile, including training status (*Perroni et al., 2023*).

The following main limitations can be considered: (i) the sample size, and; (ii) the change of direction performance may be masked with the selected tests. Regarding the sample size, it must be mentioned that, due to the nature of this study, the sample is representative of a unique population (*i.e.*, the best age-group players recruited from a regional squad to participate in a national competition). Thus, the present data report the best players selected from a regional-level championship to perform in a prestigious national tournament. Regarding the theme of changing direction, other agility tests, such as repeated sprint ability and change of direction deficit, can be used to obtain deeper insights into the players' physical performance. Therefore, future studies could include comparing regional teams to better understand whether performance determinants are different among teams, as well as longitudinal analysis to understand the effect of specific training programs on these players to overcome their main weaknesses (specifically by field position). Also, it would be interesting to integrate field-based measures with training load monitoring to understand if the non-positional differences would be maintained.

## CONCLUSIONS

Overall, no effect of field position was found in U-14 Portuguese regional team football players in a set of lower limb strength and power, linear sprint and change of direction tests, and in dynamic balance. The main variables responsible for establishing the clusters were the SJ and CMJ (lower limb strength and power) and the 30 m linear sprint. There was no cluster exclusively (or mainly) composed of players of the same field position. Thus, coaches must understand whether the lack of differentiation in physical performance between players of different positions is "normal" in regional-level young players. Otherwise, they should be aware that specific training programs for physical traits based on the players' field position may be the proper way to develop young regional-level players. These programs must be based on the individual characteristics of the players who can be grouped together based on their similarities.

## ACKNOWLEDGEMENTS

The authors are grateful to all board members, coaches and players of the Bragança Football Association for the research collaboration.

### Funding

The authors received funding from the FCT—Portuguese Foundation for Science and Technology under the project UIBD/DTP/04045/2020. The funders had no role in study design, data collection and analysis, decision to publish, or preparation of the manuscript.

## Grant Disclosures

The following grant information was disclosed by the authors:
FCT—Portuguese Foundation for Science and Technology: UIBD/DTP/04045/2020.

## Competing Interests

The authors declare that they have no competing interests.

## Author Contributions

- Tatiana Sampaio performed the experiments, analyzed the data, prepared figures and/or tables, authored or reviewed drafts of the article, and approved the final draft.
- Daniel Marinho conceived and designed the experiments, authored or reviewed drafts of the article, and approved the final draft.
- José Eduardo Teixeira performed the experiments, authored or reviewed drafts of the article, and approved the final draft.
- João Oliveira performed the experiments, prepared figures and/or tables, authored or reviewed drafts of the article, and approved the final draft.
- Jorge Morais conceived and designed the experiments, analyzed the data, prepared figures and/or tables, authored or reviewed drafts of the article, and approved the final draft.

## Human Ethics

The following information was supplied relating to ethical approvals (*i.e.*, approving body and any reference numbers):

The Polytechnic Ethics Board (Instituto Politécnico de Bragança) approved the study (No. 127/2023).

## Data Availability

The raw data are available in the Supplemental File.

## Supplemental Information

Supplemental information for this article can be found online at http://dx.doi.org/10.7717/peerj.15609#supplemental-information.

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
