# Peer review of "Clustering U-14 Portuguese regional team football players by lower limb strength, power, dynamic balance, speed and change of direction: understanding the field position factor"

_PeerJ, doi:10.7717/peerj.15609_

## Round 0.1 · original submission · Major Revisions

· Academic Editor

Major Revisions

The authors are commended for a well-written manuscript. The arguments for the manuscript under review are original and interesting. There are, however, major concerns and methodological issues with the manuscript in its current form, which need to be revised for publication. Please see the reviewers' comments below.

Reviewer 1 ·

Basic reporting

The article is interesting and presents some useful information on the way to group players when managing targeted training programs for U14 football players by coaching staff. The fact that clustering based on individual characteristics than roles appears more appropriate, it goes in the same direction of a modern interpretation of soccer performance, which should be viewed under players'function on the field rather than their mere playing position. However, there are several points that should be addressed in order to improve the readability, understanding, and potential future study replication.

Experimental design

The research question and aim are well taken. I strongly recommend the authors to justify the sample size in order to better support data interpretation.

lines 124-133: it appears that the change of direction ability may be masked with the selected tests. What do the authors think about it? As the number of turns increases (n >1), the assessment of COD ability becomes confounding, especially if the completion time remains the only variable to express it compared to other (i.e., COD deficit). It might be argued within the study limitation paragraph.
Line 137: please provide the height at which the timing gates were set, and their distance from the starting line. These data are useful from the perspective of a replication study.

line 169: why did the authors select a 45-cm height stable platform?provide a justification.

Validity of the findings

I consider the analysis and conclusions sufficiently clear.
In my opinion, the results appear well-supported by the study design and data analysis. However, the authors need to justify the sample size. Without any justification, the interpretation remains weak.

Additional comments

Abstract

lines 30-31: please use an extended name form (prior to abbreviating)
line 39: replace "must" to "should". Be more conservative.

Intro
line 56: please add a potential (more) reference supporting the sentence on the study of linear sprint and COD in youth. Here below the suggestion:

1) doi: 10.7717/peerj.9486

line 80: start the sentence as "Although differences...". Delete "Moreover, and"

M&M

lines 124-133: please add appropriate references

Discussion

line 225: remove the typo "Add"
line 229: delete "confirmed these findings, i.e., where" and continue the sentences as "Moreover, cluster analysis that any cluster was ..."
lines 289-29: please rephrase for clarity.

Reviewer 2 ·

Basic reporting

The purpose of the manuscript “Clustering U-14 Portuguese regional team football players by lower limbs strength, power, dynamic balance, speed and change of direction: understanding the field position factor” was to cluster U-14 Portuguese regional team football players based on variables related to lower limbs strength and power, dynamic balance, linear sprint and change of direction. Unfortunately, this manuscript presents many limitations and weakness. It is well written in English. Consdidering tha age of subjects there aren't considerations and refereees about pubertal developmento. You can see works of perroni and al. which dpubblished on age, pubertal and sub elite soccer player (IJERPH 2020, JSCR 2015, 2018 and 2020, Plos One 2019, Open Sports Sciences Journal 2014 and 2017).

Experimental design

In abstract I think that is it correct to insert the name of test in Methods and data value in results.
I thinks that it is correct to delete the reference of table 1 from "Partecipants" but to leave it in the Results.
There is a lack of the description of the evaluation of Anthropometrics (Height, Body weight) and pubertal (Maturity offset) characteristics in methods....please insert
Line 125 slalom test (COD) and Line 130 forward-back-forward test (COD): they have the same acronimous...To help the reader you have to change
Who are the authors of protocol of test used?....Please insert in each test

Validity of the findings

Conclusions have to insert the data value to compare with other article and to increase the a "take-home" message and practical application.

Reviewer 3 ·

Basic reporting

The manuscript is well written, although some sentences need revision as redundant words and passages are present. I advise the text to be checked by a fluent English speaker.

The literature presented is sufficiently recent and complete, despite some key papers are missing. For example, I recommend considering this recent paper, which applied factor analysis to differentiate players' roles according to a battery of physical tests, which is quite similar to the aim of the current study (https://journals.lww.com/nsca-jscr/Abstract/9900/Use_of_Exploratory_Factor_Analysis_to_Assess_the.218.aspx; doi: 10.1519/JSC.0000000000004414).

Tables and figures have sufficient quality in their current form. Nonetheless, I have a couple of additional comments: why "1 SD" is reported in the Tables and in the text, and not simply "SD" as standard reporting? It is not clear to me what the Territorial Map (Fig 2) represents: what is the meaning of 'Function 1' and 'Function 2' in the axes?

Experimental design

As already acknowledged by the authors, the main limitation of this study is the low sample size. I do not believe that 24 players could be representative of a population; as this is an unsolvable issue, authors should be more cautious when drawing conclusions based on their data.

Some important details are missing in the Methods section:
- in which period of the season were the data collected?
- please add the reliability data of the instruments used.
- the execution of the test (for example, SJ and CMJ tests) needs to be explained more clearly.
- what number of clusters was chosen, and why? Usually, statistical methods (such as elbow, silhouette, and gap methods) are applied to determine the best number of clusters, it should not be an arbitrary decision.
- alpha value used was not present, nor was the software used for the analysis.
- it is not clear whether the goalkeepers were excluded from the analyses.

What do the authors mean by "a non-significant field position was noted"? I would state it more clearly, saying that "no effect of field position was found/detected."

Validity of the findings

I have concerns regarding the validity of the results in terms of applicability to a wider population. Authors stated that the players were "the best selectable to perform in a well prestigious National tournament". I would say that they were the best (selected according to which criteria?) players of a specific region of Portugal, but this nothing says about their level: so, authors should be cautious when referring to them as "sub-elite".

Additional comments

The use of the word "handicap" should be avoided in this context, as it is a non-inclusive language. Please, find a synonym to use: maybe 'weaknesses' or 'deficiencies' could work?

Please, pay attention to the use of acronyms: some acronyms are present (see the abstract for example) without explicitly the whole word on first use.

In the Participants section, it is not clear what "-0.93 ± 0.80 years" means.

---

## Round 0.2 · accepted · Accept

· Academic Editor

Accept

Dear Author. Thank you for addressing all of the reviewers' comments. The current version of the manuscript is, in my view, ready for publication. Congratulations.